# The Impact of Childhood Trauma on Problematic Alcohol and Drug Use Trajectories and the Moderating Role of Social Support

**DOI:** 10.3390/ijerph20042829

**Published:** 2023-02-06

**Authors:** Christopher J. Rogers, Myriam Forster, Steven Sussman, Jane Steinberg, Jessica L. Barrington-Trimis, Timothy J. Grigsby, Jennifer B. Unger

**Affiliations:** 1Department of Health Sciences, California State University, Northridge, CA 91330, USA; 2Department of Population and Public Health Sciences, Keck School of Medicine, University of Southern California, Los Angeles, CA 90007, USA; 3Department of Environmental and Occupational Health, University of Nevada, Las Vegas, NV 89154, USA

**Keywords:** adverse childhood experiences, social support, alcohol, drug use

## Abstract

Adverse childhood experiences (ACE) have a strong association with alcohol and drug use; however, more research is needed to identify protective factors for this association. The present study assesses the longitudinal impact of ACE on problematic alcohol and drug use and the potential moderating effect of perceived social support. Data (n = 1404) are from a sample of Hispanic youth surveyed in high school through young adulthood. Linear growth curve models assessed the effect of ACE and perceived social support over time on problematic alcohol and drug use. Results indicated youth with ACE (vs. those without ACE) report more problematic alcohol and drug use in adolescence and have increased rates into young adulthood. Additionally, findings suggest that social support in high school may moderate the effects of ACE on problematic use over time. Among youth with high levels of support, the association of ACE with problematic alcohol and drug use was diminished. Although ACE can have a persistent impact on problematic alcohol and drug use from adolescence into adulthood, high social support during adolescence may mitigate the negative effects of ACE, lowering early problematic alcohol and drug use, offering the potential for lasting benefits.

## 1. Introduction

The annual health, crime, and lost productivity costs associated with substance use disorders in the United States (US) are estimated to be 740 billion USD [1,2,3]. Patterns of alcohol and drug use and subsequent negative outcomes across the life course are heterogeneous in the U.S. population; nevertheless, alcohol and drug misuse is a major public health concern [4]. For a subset of users, functioning (socially or physically) diminishes over time as a result of substance use behaviors [5] and is associated with a spectrum of negative consequences [5]. This problematic use of alcohol or drugs often persists across the lifespan with consequences that include premature mortality, morbidity, criminality, and lost productivity that take a toll on the individual and society. Studies of longitudinal patterns of problematic substance use and treatment have identified cycles of cessation and relapse; however, compared to cross-sectional research, there is far less longitudinal evidence. Additionally, much of the conducted cohort research has focused on frequency and patterns of use rather than risk and protective factors [4]. Reviews assessing consequences associated with problematic use suggest that continued research would improve our understanding of factors that increase harmful consequences and those that can limit negative outcomes [6,7,8,9,10]. This highlights the need for more research into risk and protective factors related to problematic drug and alcohol use [11,12].

Adverse childhood experiences (ACE) are negative events occurring before the age of 18 conceptualized as child maltreatment (e.g., sexual, physical, and verbal abuse) and household dysfunction (e.g., parental divorce, substance use and household mental illness, incarceration, and homelessness) [13]. ACE, a set of highly correlated events, are consistent predictors of poor health outcomes, including dysregulated engagement in addictive behaviors [14,15,16,17,18]. Individuals with a history of ACE tend to have significantly more life course health problems and negative outcomes known to contribute to premature morbidity and mortality [14]. ACE can disrupt physiological pathways during development, result in cognitive and emotional impairment, and strain individuals’ coping capacity [14,19,20,21], exacerbating the risk for maladaptive coping behaviors. Trauma-related cognitive and emotional deficits contribute to emotional dysregulation and disrupted attachment, which increase vulnerability for maladaptive coping behaviors, especially addictive behaviors that can temporarily limit feelings of distress [22,23,24,25,26]. Since the seminal study introducing the adverse childhood experiences (ACE) framework by the Kaiser Family Foundation [13], there has been a growing interest in the impact of ACE on behavioral health. ACE have been associated with alcohol and drug use frequency as well as alcohol- or drug-related negative consequences [27,28,29,30,31,32,33]. Given the importance of the life course perspective and the periods of adolescence and young adulthood in relation to addiction, research that advances our understanding of problematic alcohol and drug use and the impact of ACE longitudinally from adolescence through young adulthood is critical, especially among this vulnerable subset of the population.

From a prevention and treatment perspective, identifying resilience and mediating factors that can buffer the ACE–addiction relationships is paramount for young people with and without internalizing symptoms resulting from trauma exposure [34]. Social support, psychological and material resources to assist in an individual’s capacity to cope with stress [35,36], has been identified as a positive factor in mental and physical health outcomes and one that can mitigate the effects of trauma and stress [37,38,39,40]. Social support can decrease maladaptive coping and promote behaviors that improve stress regulation, increase confidence, decrease engagement in risky behaviors, and promote healthy and effective coping strategies [39], which are all important for addiction prevention [41]. For adolescents specifically, social support can limit dysregulation in the face of traumatic stressors [39,42] and promote positive mental health outcomes, even for ACE-exposed individuals [43]. Despite evidence of the benefits of social support among trauma-exposed populations, of the stress buffering effects of social support [44,45,46,47], and of social support being a key ingredient of resilience among adults [48,49], there is limited research assessing social support in the context of ACE and behaviors with addictive potential. As such, ongoing research with adolescent and young adult populations is needed to investigate the potential buffering role of social support in the relationship between ACE and problematic alcohol and drug use among adolescents longitudinally through young adulthood. The current study explored the patterns of ACE and alcohol and drug use consequence trajectories over time.

### Study Aims

This study examined (1) the impact of ACE in trajectories of past 30-day problematic alcohol and drug use from adolescence to young adulthood and (2) the potential protective effects of perceived social support in the ACE—problematic alcohol and drug use trajectories from adolescence to young adulthood. We hypothesize that (H1) at baseline, higher ACE will be associated with higher problematic alcohol and drug use and higher perceived social support will be associated with lower problematic alcohol and drug use, (H2) higher ACE exposure in young adults will be associated with a greater increase in problematic alcohol and drug use over time compared to young adults who were not ACE exposed, and (H3) the differences in trajectories of problematic alcohol and drug use over time by ACE exposure will be moderated by perceived social support; for example, those with higher adolescent perceived social support will have similar trajectories as non-ACE-exposed students, in contrast to those with low perceived social support.

## 2. Materials and Methods

### 2.1. Participants and Procedures

Participant information was derived from Project Reteniendo y Entendiendo Diversidad para Salud (RED), a longitudinal cohort study designed to assess acculturation and substance use patterns among Hispanic/Latino adolescents enrolled public high schools in Southern California [50,51]. Adolescents who were initially enrolled attended one of the eight randomly selected high schools in the Los Angeles area with student bodies that were at least 75% Hispanic (as indicated by data from the California Board of Education). Investigators visited classrooms of all eight high schools and distributed parental consent and youth assent forms. The first survey wave occurred in 2005 with 9th grade students and then repeated in 10th and 11th grade. To extend the study beyond high school into young adulthood, the cohort was re-contacted in 2011. This study led to five post-high school waves of data collection that occurred in 2011, 2013, 2014, 2016, and 2018. The Institutional Review Board (IRB) approved all study procedures. Problematic alcohol and drug use was measured in the final high school survey and in the first four young adulthood surveys. The sample was restricted to (1) only respondents who provided data on ACE (n = 1404), collected retrospectively in 2013; (2) those who used (lifetime) alcohol or (lifetime) drugs at some point in any wave; and (3) provided problematic alcohol and drug use data on at least one of the surveys where it was collected (2009, 2011, 2013, 2014, and 2016). Some of the original cohort (n= 3218) was lost to follow-up, with the final analytic sample composed of 1404 participants with data from six survey waves.

### 2.2. Measures—Dependent Variables

Self-reported problematic alcohol and drug use was assessed with a modified version of the Rutgers Alcohol Problem Index [10], adapted for general drug use [52,53]. Seven items were prefaced with the following statement, “Different things happen to people while they are drinking alcohol or using other drugs or because of their alcohol drinking or use of other drugs. How many times has each of these things happened within the last month due to drinking or drug use?” Questions included, for example, “not able to do your work or study for a test?”, “got into fights with other people (friends, relatives, strangers)?”, and “felt physically or psychologically dependent on alcohol or drugs?”. Response options included 1 = “never”, 2 = “sometimes”, 3 = “often”, and 4 = “more than five times”. The final items were summed. The questions were asked in the final year of high school (survey 3) and then the first four post-high school surveys (Survey 4–7). The final variable ranged from (7 to 28), with Cronbach’s alpha showing good internal consistency (α = 0.820).

### 2.3. Measures—Independent Variables

Time was a continuous variable accounted for in the model. Nine self-reported adverse childhood experiences were assessed with items measured retrospectively in the second post-high school wave of data collection. Items assessed child maltreatment (e.g., physical, sexual, and verbal abuse) and household dysfunction (e.g., parental partner violence, incarceration, alcohol misuse, illicit substance use, mental illness, and divorce). The maltreatment questions were prefaced with, “While you were growing up, that is your first 18 years of life, did a parent, step-parent, or adult living in your home often…” in line with the original ACE measure [13]. Response options were coded 1 = “yes” and 0 = “no”. Physical abuse was assessed with two items and endorsement of abuse was coded with a report of yes to either question: “Push, grab, slap, or throw something at you?” or “Hit you so hard that you had marks or were injured?” Verbal abuse was assessed with two items and endorsement of abuse was coded with a report of yes to either question: “Swear at you, insult you, or put you down?” or “Threaten to hit you or throw something at you, but didn’t do it?” Sexual abuse was assessed with four items and endorsement of abuse was coded with a report of yes to either question: “Touch or fondle your body in a sexual way?” or “Have you touched their body in a sexual way?” or “Attempt to have any type of sexual intercourse with you (oral, anal, or vaginal)?” or “Actually, have any type of sexual intercourse with you (oral, anal, or vaginal)?” Household disfunction items asked participants if, before they turned 18 years old, they lived with anyone who was mentally ill, misused substances, was incarcerated, or was physically violent with their spouse/partner. Response options were 1 = “yes”, or 0 = “no”. Parental relationship violence was assessed with four items and endorsement of abuse was coded with a report of yes to either question asking if their mother was: “Pushed, grabbed, slapped, or had something thrown at her?” or “Kicked, bitten, hit with a fist, or hit with something hard?”, “Repeatedly hit over at least a few minutes?”, or “Threatened with a knife or gun, or if someone used a knife or gun to hurt her?” Household mental illness was assessed with two items and endorsement of illness was coded with a report of yes to either question: “Was anyone in your household depressed or mentally ill?” or “Did anyone in your household attempt to commit suicide?” The final four household dysfunction items assessed household problematic alcohol use, household drug use, household divorce, and household incarceration. Affirmative responses to ACE items were summed to create an index of childhood adversity (range 0–9).

### 2.4. Measures—Moderator

*Perceived Social Support* was assessed with the Multidimensional Scale of Perceived Social Support scale [54]. The questions were asked at baseline data in the first survey. Response options included 1 = “strongly disagree”, 2 = “somewhat disagree”, 3 = “agree”, and 4 = “strongly agree”. There was a total of 12 items, with 4 items about support from a “special person”, 4 items about support from family, and 4 items about support from friends (Cronbach’s Alpha = 0.902).

### 2.5. Measures—Covariates

Demographic covariates included self-reported sex (0 = female, 1 = male), nativity (0 = Other, 1 = U.S. born), and socioeconomic status. This study created a standardized index to represent SES that has been validated in this population [51,55]. The index included parent’s education rated on a six-point scale ranging from ‘‘8th grade or less’’ to “advanced degree”; a ratio of the number of rooms per person in the home captured by dividing the number of people in the house by the number of rooms in the house, and the U.S. census median household income in the respondent’s provided zip code. The index also included dichotomous measures of eligibility for free/reduced price lunch at school (1 = no, 0 = yes), homeownership (1 = family owns its home, 0 = family rents home from a landlord), presence of a computer in the home (1 = yes, 0 = no), presence of a gaming console in the home (1 = yes, 0 = no), and availability of the Internet at home (1 = yes, 0 = no). To weight each indicator equally, items were standardized to a mean of 0 and a standard deviation of 1 and summed [51,55]. Higher values have more indicators of higher SES. Dummy-coded variables for schools were included to control for any school level differences.

### 2.6. Statistical Analysis

Mean plots were calculated to graph the problematic alcohol and drug use score trends across time. Random linear growth curve models, using SAS PROC MIXED, estimated the problematic alcohol and drug use trends across time using random slopes, random intercepts, and the fixed effects of ACE and demographic covariates. Random effects for schools were not included because of low intraclass correlation coefficients (ICC) of students nested within schools (<0.03). Models were iteratively assessed to determine appropriateness of model parameters beginning with empty models and subsequently adding random effects of time and differing covariance patterns. The models with the best fit were random linear models that included the random intercepts and slopes for time, as well as the fixed effects of time invariant independent variable (ACE), and the time invariant baseline model covariates (SES, sex, nativity, and school). The inclusion of the fixed effects of ACE and social support were used to test hypothesis 1. After the final best fit models were assessed, interaction terms were included to assess the cross-level effect of moderators with time on problematic alcohol and drug use. To test hypothesis 2, the first model included an interaction term (ACE × time) to determine the cross-level effect of ACE with time on problematic alcohol and drug use. To test hypothesis 3, the second model included a three-way interaction with the ACE × time × perceived social support, along with all covariates and lower order interaction terms, to determine if there were differences in the ACE × time interaction by perceived social support. This approach can identify differences in the trajectories of average problematic alcohol and drug use scores over time, whether these are potentially exacerbated by ACE, and if they are moderated by early social support in high school. To visualize interaction effects, the final model’s predicted values were stratified by levels of ACE and plotted with 95% confidence intervals using the SAS PROC PLM procedure with panels of different perceived social support levels from low to high support. To address missing data across waves, maximum likelihood estimation of mixed models allows for data that are missing the dependent variable [56,57]. Therefore, the sample was restricted to participants with complete data on ACE and alcohol or any substance across any survey wave, but still allowed for missing data on other variables. All statistical tests were performed using SAS v9.4.

## 3. Results

The final analytic sample was comprised of 1404 Hispanic participants who provided data on ACE and problematic alcohol and drug use on at least one time point. For this analytic sample, 88% were U.S. born and just over half (59%) were female (Table 1). At baseline, the average age was 16.47 (SD = 0.39) and at the fourth wave, the average age was 23.87 (SD = 0.42). On average, participants reported 2.75 (SD = 2.19) ACE with 58% of the sample reporting verbal abuse, 51% physical abuse, 37% parental divorce, 30% household alcohol use, 25% parental intimate partner violence, 22% household mental illness, 22% household incarceration, 17% household drug use, and 16% sexual abuse.

The mean past 30-day problematic alcohol and drug use score was 8.3 (SD = 2.2) at the third high school wave, 7.9 (SD = 2.3) at the first young adulthood wave, 8.1 (SD = 2.6) at the second young adulthood wave, 8.1 (SD = 2.3) at the third young adulthood wave, and 8.4 (SD = 2.5) at the fourth young adulthood wave. Growth curve analysis estimated participants’ trajectory of problematic alcohol and drug use scores from the final high school year through emerging adulthood, with five waves of data collection (Table 2).

Across all models, there was significant variance in random slopes; however, there was only significant variance in random intercepts for models 1 and 2. Based on model 1, when assessing the fixed linear effect of ACE on substance use at baseline, ACE was a significant predictor for differences in problematic alcohol and drug use scores. Those with higher ACE had significantly higher problematic alcohol and drug use scores at baseline (β = 0.159, 95% CI = 0.11, 0.21). When assessing the fixed linear effect of perceived social support in high school on problematic alcohol and drug use scores at baseline, those with higher support had significantly lower levels of problematic alcohol and drug use at baseline (β = −0.025, 95% CI = −0.05, −0.01) supporting H1. Sex was also significant; at baseline, males had higher levels of problematic alcohol and drug use than females (β = 0.514, 95% CI = 0.33, 0.69).

In model 1 (Table 2), the interaction term between time and ACE was positive and significant, indicating that the linear rate of change over time differed across levels of ACE for problematic alcohol and drug use (β = 0.02, 95% CI = 0.01, 0.03). At baseline, those with higher ACE start with more problematic alcohol and drug use and then continue to increase at a greater rate than those with lower levels of ACE. At higher levels of ACE, there is a steeper trajectory of problematic alcohol and drug use scores, meaning that on average, over time, those with higher ACE will experience more problematic alcohol and drug use than respondents with lower ACE (Figure 1). In model 1, perceived social support is also significant, indicating that at baseline, youth with higher perceived social support have lower problematic alcohol and drug use.

In model 2, the three-way interaction term between time and ACE and perceived social support was positive and significant, indicating that the linear rate of change over time differs across levels of ACE for problematic alcohol and drug use and that this relationship operates differently across levels of perceived social support. This finding suggests that support may moderate the effects of ACE in problematic use over time (β = 0.005, 95% CI = 0.001, 0.009). Across all levels of perceived support, youth with higher ACE start with higher problematic alcohol and drug use scores; however, the effect of ACE is much more pronounced among individuals with less support. Respondents who had low levels of perceived social support in high school and high ACE had more problematic substance use at baseline than those with more perceived social support, regardless of ACE level. There were also changes over time across ACE and support. Although respondents with the lowest support and high ACE start with the highest problematic use scores, all ACE groups begin to regress to the mean over time; however, respondents with high ACE still have higher substance use than those with no ACE. For participants with the highest level of support, although all ACE levels begin at a lower level, those with more ACE appear to increase in problematic alcohol and drug use over time, and those with lower ACE appear to decrease in problematic alcohol and drug use over time. It is important to note that participants with the highest level of ACE and the highest level of support in adolescence will have lower problematic alcohol and drug use in young adulthood compared to their peers with the lowest support and higher levels of ACE (Figure 2).

## 4. Discussion

This study assessed the impact of ACE in problematic alcohol and drug use longitudinally from adolescence through young adulthood and explored the potential moderating role of high school perceived social support. Epidemiologic evidence suggests that the prevalence of drug use increases over the course of adolescence and peaks in young adulthood and that understanding patterns of problematic use across this transition period are necessary; however, there is limited longitudinal research assessing these trajectories within this life period [58], despite the identification of important developmental risk factors, such as childhood exposure to adversity. The results of the current study contribute to this body of evidence and indicate that the effects of ACE on problematic substance use can persist over time and that perceived social support during high school may attenuate the effect of ACE for problematic use. Early life course substance use can lead to escalating substance use and may impact academic performance, professional development, and general health and wellbeing across the life course, making it a critical public health priority.

There was evidence supporting hypothesis one, i.e., that every additional ACE at baseline was associated with higher problematic alcohol and drug use scores in high school. There was also evidence supporting hypothesis two, i.e., that adolescents with higher ACE exposure had a greater increase in problematic alcohol and drug use into young adulthood compared to those who were not ACE exposed. Adolescent use of alcohol and drugs is a key indicator for problematic use later in life, and early problematic use has been associated with negative developmental and psychological differences in youth [59]. This can lead to continued or worsening consequences as youth transition to young adulthood, where they will have more autonomy and will make many critical life decisions. The cross-level effects of ACE illustrate that youth with higher ACE have differential trajectories through young adulthood compared to youth with lower ACE or no ACE. Specifically, not only do ACE-exposed individuals start with higher problematic alcohol and drug use scores, but problematic use continues over time while their peers with low to no ACE, who start lower at baseline, appear to decrease their problematic use over time. Youth and young adults with high ACE exposure have an increasing likelihood of future problematic alcohol and drug use highlighting the potential persistent impact of ACE on problematic use. In contrast, respondents with no history of ACE appear to have a decreasing likelihood of young adult problematic alcohol and drug use.

Given the persistence of the effect of ACE over time, it is important to identify potential protective factors that can reduce problematic alcohol and drug use or attenuate escalating use over time. We identified that strong perceived social support in high school, as early as 9th grade, was associated with lower problematic alcohol and drug use at baseline and found evidence for hypothesis three, i.e., that the differences in trajectories of problematic alcohol and drug use over time by ACE exposure were moderated by perceived social support. At very low levels of perceived social support, there are large differences in problematic substance use across ACE. Students with high ACE have much higher problematic use scores at baseline, and over time, problematic use does diminish some, but among lower ACE-experiencing individuals, problematic use scores increase with all groups regressing toward the mean; however, those with higher ACE are still significantly higher at wave 5 compared to non-ACE-exposed individuals. In other words, low perceived social support may exacerbate the effects of ACE for problematic use of alcohol and drugs, creating even greater disparities at baseline between those who are ACE- and non-ACE-exposed. The changes over time for this low support group suggest that even the non-ACE-exposed individuals with limited perceived social support may eventually be at risk for problematic use in high school. In contrast, at very high levels of perceived social support, there are few baseline differences in problematic alcohol and drug use scores (everyone starts much lower), which over time, are lower than individuals with low levels of perceived social support. However, it is important to note that the increases in problematic use among ACE-exposed individuals are still much lower for those with support compared to those with low support. Both the baseline and over time differences across levels of support demonstrate that fostering social support in high school may be an intervention and prevention opportunity for programs targeting adolescents and young adults.

In sum, these findings underscore the importance of intervention and prevention research and services for youth with ACE. Given the prevalence of ACE and the cost of associated health issues, providing adolescents and young adults the tools, training, and support necessary to manage traumatic stress, as well as encouraging the development of positive relationships to cope with their trauma, should be a public health priority. These results are in line with other research that has shown that cultural assets and support systems may in fact mitigate the negative effects of ACE for youth and young adult health and well-being [60,61,62,63,64].

Social support can moderate the ACE-problematic use relationship; however, future research that examines social support across adolescence and young adulthood would add to our understanding of the timing of effects. Studies using longitudinal designs that assess support at different time points would help identify key periods at which support interventions could reduce ACE-related negative patterns of substance use behaviors. Future research should also consider examining differences in the sources and types of support people receive. This may be critical given that support from school staff or friends is likely to operate differently from family and community. Understanding these differences and benefits can improve the effectiveness of prevention efforts for young people.

### Limitations

First, ACE and substance use data were self-reported; however, self-reported substance use has been found to be highly accurate under confidential survey conditions such as the present study [65]. Second, ACE was assessed retrospectively. However, much of the ACE research has been conducted retrospectively and even studies challenging retrospective versus prospective reports suggest that retrospective reports provide a meaningful addition to the literature and are validly associated with other subjective measures [66]. Third, we cannot definitively anchor ACE to a specific time-point in childhood; however, the survey specifically refers to events that occurred prior to age 18. Similarly, we were not able to assess whether the timing of ACE occurred prior to any high school trajectory changes or prior to substance use initiation. Fourth, our assessment of ACE and problematic alcohol and drug use were not comprehensive and may underestimate the burden of childhood adversity and problematic use in this sample. Fourth, similar to other longitudinal studies, there was a considerable amount of attrition, and the study used the maximum likelihood estimations built into the SAS PROC MIXED procedure which produces unbiased parameter estimates by accounting for all included data and other model covariates, even with missing data on the dependent variable [56,57]. Fifth, participants that were excluded due to attrition or not providing information on ACE and substance use may represent an especially vulnerable subset of the sample, such that the results may only provide a preliminary understanding of these relationships. Sixth, due to the limited variability in age, it was not included as a covariate (all students entered the study as first-year students). Race was also not included because the sample was restricted to students who identified as Hispanic. Future studies further disaggregating Hispanic groups or further study into other racial/ethnic population may be needed. Finally, the sampling design restricted the sample to Hispanic/Latinx individuals. This does provide needed information in a critical minority population; however, it is not generalizable to all populations.

## 5. Conclusions

ACE can have a persistent impact on problematic alcohol and drug use, as seen in the trajectories of youth with a history of co-occurring ACE. Compared to youth who experience fewer or no ACE, this vulnerable subset of students is at greater risk for problematic use as early as age 14 that persists through young adulthood. However, and importantly for prevention and intervention efforts, the moderating effect of perceived social support during high school can attenuate the effect of ACE for problematic use. This early reduction of problematic alcohol and drug use may present a lasting benefit and limit the negative effects of ACE and the impact of problematic use. The results of this study emphasize the potential benefits of social support during high school in reducing problematic alcohol and drug use and attenuating the impacts of ACE on these relationships, disrupting the risk trajectories among Hispanic youth and emerging adults. Meaningful and supportive personal relationships have tremendous value and it may be advantageous to promote relationship building via campus programs, mentorships, or other interventions.

## Figures and Tables

**Figure 1 ijerph-20-02829-f001:**
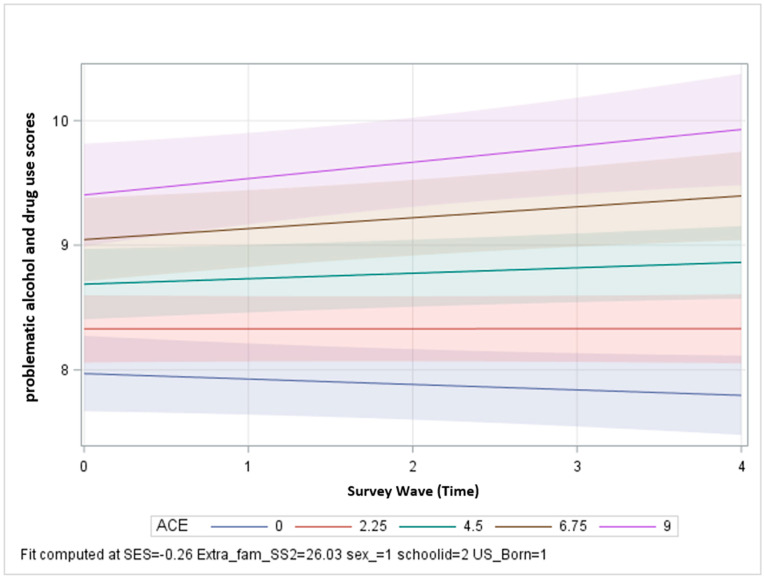
Problematic alcohol and drug use scores over time by level of ACE exposure, controlling for covariates.

**Figure 2 ijerph-20-02829-f002:**
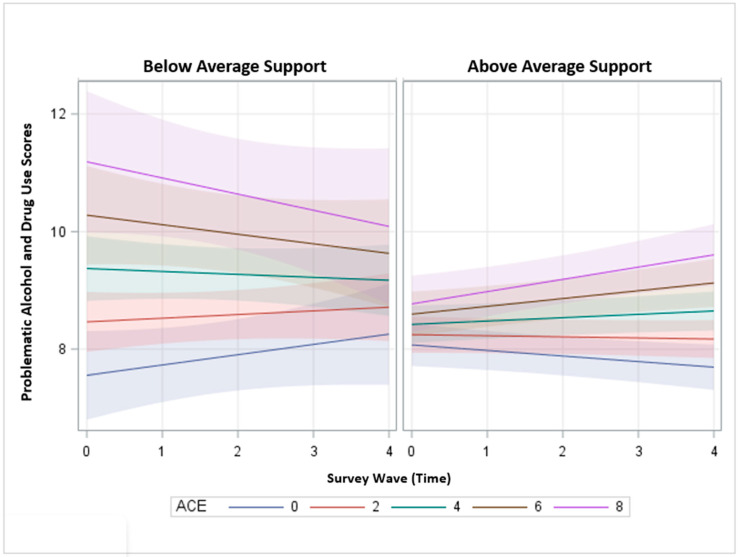
Problematic alcohol and drug use scores over time by level of ACE exposure, controlling for covariates and paneled by perceived social support.

**Table 1 ijerph-20-02829-t001:** Non time-dependent descriptive statistics (N = 1404 participants).

Variable	Frequency	Percent
Nativity		
U.S. Born	1228	87.78%
Not U.S. Born	171	12.22%
Sex		
Male	570	40.60%
Female	834	59.40%
**Variable**	**Mean**	**Std Dev**
Adverse Childhood Experiences	2.75	2.19
Perceived Social Support (in adolescence)	26.02	4.01
Socioeconomic Status (in adolescence)	−0.28	3.70

Notes: Std Dev = Standard Deviation.

**Table 2 ijerph-20-02829-t002:** Results of growth curve models.

	MODEL 1 Problematic Alcohol and Drug Use—ACE × Time Interaction Model	MODEL 2 Problematic Alcohol and Drug Use—ACE × Time × Support Model
Variance Components	Parameter Estimate (SE)	Parameter Estimate (SE)
UN1,1	1.830 *** (0.1678)	1.803 *** (0.167)
UN2,1	−0.1483 ** (0.0513)	−0.141 ** (0.051)
UN2,2	0.1521 *** (0.023)	0.151 *** (0.023)
Time	3.357 *** (0.080)	3.359 *** (0.081)
Fixed Effects	Parameter Estimate (SE)	Parameter Estimate (SE)
Intercept	7.995 ** (0.353)	6.6568 (0.623)
Time	−0.044 (0.032)	0.309 (0.216)
ACE	0.159 *** (0.026)	0.637 *** (0.173)
Support	−0.025 * (0.011)	0.025 (0.226)
Sex	0.514 *** (0.092)	0.513 *** (0.092)
Time × ACE	0.019 * (0.009)	−0.103 (0.061)
Time × Support		−0.018 (0.008)
ACE × Support		−0.018 ** (0.006)
Time × ACE × Support		0.005 * (0.002)

Notes: *p* < 0.05 *, *p* < 0.01 **, *p* < 0.001 *** The models also controlled for SES, nativity, and school.

## Data Availability

Data are available for download through the Inter-university Consortium for Political and Social Research. Unger, Jennifer. Drug Use and Cultural Factors Among Hispanic Adolescents and Emerging Adults, Los Angeles, 2006–2016. Inter-university Consortium for Political and Social Research [distributor], 2018-10-03. https://doi.org/10.3886/ICPSR36765.v2 (accessed on 15 November 2022).

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
