# Peer review of "The Impact of Childhood Trauma on Problematic Alcohol and Drug Use Trajectories and the Moderating Role of Social Support"

_ijerph, 2023, doi:10.3390/ijerph20042829_

Round 1

Reviewer 1 Report

 In the Methodology I suggest that 

It would provide a better insight for the reader if the authors could describe their methods separately in testing each of their hypotheses.

Author Response

We would like to thank the reviewers and the editors for the opportunity to resubmit the manuscript. We have addressed the comments both in this response and in the text. Thank you so much for your time and careful review of the manuscript.

Reviewer 1

1-It would provide a better insight for the reader if the authors could describe their methods separately in testing each of their hypotheses.

Thank you for this opportunity to clarify the methods. We have now edited the analysis section to include explanation of what parts are for each of the hypothesis.

Reviewer 2 Report

This fascinating paper reports on a longitudinal study of relationships between experiencing childhood adversity (ACE) and trajectories of problematic alcohol and drug use between 9th grade (age 16) and age 24 among a large sample of Hispanic youth in the Los Angeles area. The primary question addressed herein is whether social support in mid-adolescence can impact the shape of these trajectories. This is an important issue, because social support is modifiable through interventions at the personal, family and community levels.

The introduction does an excellent job of setting the stage by clearly laying out the existing literature and identifying gaps in our understanding, especially the dearth of related longitudinal studies. The methods were appropriate, the sample was likely representative of the population of interest, and the measures were widely used in other research and demonstrated good psychometric properties with this sample. The analytic approach was highly sophisticated and well executed.

I am impressed with how clearly the results were presented. Kudos on the figures! They illustrate very clearly the complex relationships among ACE, problematic SU, and support. Additionally, the text teased out the very complex relationships in a manner that was easy to grasp. I didn't have to read passages repeatedly to understand your points. Thank you.

The discussion also does an excellent job of analyzing the implications of the findings for future research and intervention. The point you made (lines 355-363) about studying the effects of sources of social support is critical. Too often, teens turn to gangs and other potentially harmful sources of support and increasing availability and acceptability of alternative community-based support systems should be a priority for policy makers. Research on effective models and implementation strategies is key.

Limitations are discussed fully, and the approaches you took to minimize them are laudable.

My only suggestions for revisions are related to minor errors:

Dr. Grigsby's email address was substituted for Dr. Unger's on the title page (listing under CSUN).

 Lines 144-146: There appears to be an error in wording. The stem is described as "... how often did a parent ...?" But Table 1 in the original citation for the ACE measure (#54) describes the wording of the stem as "... did a parent ... often ...?" The latter wording allows binary coding as no/yes without translation from an ordinal scale. Please correct.

Line 163: "... “Kicked, [bitted], hit with a fist..." Should the bracketed word be "bitten" or "butted"?

Line 180: You describe querying self-reported gender, with only male/female response options. Gender is far more nuanced, requiring a range of non-binary options in addition to male and female. I think you actually measured self-reported sex. Please change this in text and Table 1.

Table 2: I don't understand the meaning of "In" on the line above the estimates for support. Is that a typo or should it be located elsewhere and explained?

Lines 247-256: I appreciate your reporting CIs for these estimates. However, because CIs are calculated around the unstandardized coefficient (i.e., B rather than beta) it would help to report both B and beta (i.e., B = X.XX, 95% CI: Y.yy, Z.zz; beta = A.AA). That way, the negative beta and positive CI on line 251 won't look like an error.

Lines 298 & 299: "Results of the current study contributes ... and indicates ..." 'Results' is a plural noun; the verbs should match (i.e., contribute; indicate). Otherwise, you could say that the study contributes and indicates.

Lines 381-383: "Fifth, participants that were excluded due to attrition or not providing information on ACE and substance use." This is a sentence fragment. I suggest merging it with the following statement (lines 383-384).

Author Response

We would like to thank the reviewers and the editors for the opportunity to resubmit the manuscript. We have addressed the comments both in this response and in the text. Thank you so much for your time and careful review of the manuscript.

Reviewer 2

2-Dr. Grigsby's email address was substituted for Dr. Unger's on the title page (listing under CSUN).

Thank you for identifying this issue. In the version that was resent to us, it appears this issue has been corrected.

3-Lines 144-146: There appears to be an error in wording. The stem is described as "... how often did a parent ...?" But Table 1 in the original citation for the ACE measure (#54) describes the wording of the stem as "... did a parent ... often ...?" The latter wording allows binary coding as no/yes without translation from an ordinal scale. Please correct.

We acknowledge that the wording was not in line with the response options. The reviewer was correct. It was changed to the latter because that was what was asked and what fits the binary options.

4-Line 163: "... “Kicked, [bitted], hit with a fist..." Should the bracketed word be "bitten" or "butted"?

Thank you for catching this needed correction. The word was supposed to be bitten and it has now been changed.

5-Line 180: You describe querying self-reported gender, with only male/female response options. Gender is far more nuanced, requiring a range of non-binary options in addition to male and female. I think you actually measured self-reported sex. Please change this in text and Table 1.

This is a great suggestion and clarification. We have now changed the word gender to self-reported sex in the paper.

6-Table 2: I don't understand the meaning of "In" on the line above the estimates for support. Is that a typo or should it be located elsewhere and explained?

This was a typo and was not supposed to be in the table. It has now been removed.

7-Lines 247-256: I appreciate your reporting CIs for these estimates. However, because CIs are calculated around the unstandardized coefficient (i.e., B rather than beta) it would help to report both B and beta (i.e., B = X.XX, 95% CI: Y.yy, Z.zz; beta = A.AA). That way, the negative beta and positive CI on line 251 won't look like an error.

We went back to the model results to check and the unstandardized coefficients were used. The error was in the reporting of the CI. It the CI in question was supposed to be negative and the negative signs just were omitted on accident. Thank you for catching this error.

8-Lines 298 & 299: "Results of the current study contributes ... and indicates ..." 'Results' is a plural noun; the verbs should match (i.e., contribute; indicate). Otherwise, you could say that the study contributes and indicates.

Thank you for identifying this disagreement. We have changed the words so that there is agreement.

9-Lines 381-383: "Fifth, participants that were excluded due to attrition or not providing information on ACE and substance use." This is a sentence fragment. I suggest merging it with the following statement (lines 383-384).

The sentences were joined and corrected so that they can be combined. Thank you for catching this issue.